# LINE1-Mediated Reverse Transcription and Genomic Integration of SARS-CoV-2 mRNA Detected in Virus-Infected but Not in Viral mRNA-Transfected Cells

**DOI:** 10.3390/v15030629

**Published:** 2023-02-25

**Authors:** Liguo Zhang, Punam Bisht, Anthony Flamier, M. Inmaculada Barrasa, Max Friesen, Alexsia Richards, Stephen H. Hughes, Rudolf Jaenisch

**Affiliations:** 1Whitehead Institute for Biomedical Research, Cambridge, MA 02142, USA; 2HIV Dynamics and Replication Program, Center for Cancer Research, National Cancer Institute, Frederick, MD 21702, USA; 3Department of Biology, Massachusetts Institute of Technology, Cambridge, MA 02142, USA

**Keywords:** SARS-CoV-2, LINE1, retrotransposition, WGS, enrichment sequencing, RNA transfection

## Abstract

SARS-CoV-2 sequences can be reverse-transcribed and integrated into the genomes of virus-infected cells by a LINE1-mediated retrotransposition mechanism. Whole-genome sequencing (WGS) methods detected retrotransposed SARS-CoV-2 subgenomic sequences in virus-infected cells overexpressing LINE1, while an enrichment method (TagMap) identified retrotranspositions in cells that did not overexpress LINE1. LINE1 overexpression increased retrotranspositions about 1000-fold as compared to non-overexpressing cells. Nanopore WGS can directly recover retrotransposed viral and flanking host sequences, but its sensitivity depends on the depth of sequencing (a typical 20-fold sequencing depth would only examine 10 diploid cell equivalents). In contrast, TagMap enriches the host–virus junctions and can interrogate up to 20,000 cells and is able to detect rare viral retrotranspositions in LINE1 non-overexpressing cells. Although Nanopore WGS is 10–20-fold more sensitive per tested cell, TagMap can interrogate 1000–2000-fold more cells and, therefore, can identify infrequent retrotranspositions. When comparing SARS-CoV-2 infection and viral nucleocapsid mRNA transfection by TagMap, retrotransposed SARS-CoV-2 sequences were only detected in infected but not in transfected cells. Retrotransposition in virus-infected cells, in contrast to transfected cells, may be facilitated because virus infection, in contrast to viral RNA transfection, results in significantly higher viral RNA levels and stimulates LINE1 expression by causing cellular stress.

## 1. Introduction

LINE1 is the only autonomous non-LTR retrotransposon in the human genome and encompasses ~100 functional, full-length copies [1]. LINE1 retrotransposition contributes to human genetic diversity and can cause disease [1]. LINE1 is normally repressed in the majority of somatic cell types but can be de-repressed in senescent and ageing cells [2,3,4], in cancer tissues [1,5,6], or upon viral infection [7,8,9,10,11], possibly due to disrupted cell heterochromatin maintenance [7,12]. Full-length LINE1 encodes two proteins essential for LINE1 retrotransposition: L1ORF1p and L1ORF2p. L1ORF1p is a nucleic acid chaperone with high RNA-binding affinity [13]. L1ORF2p harbors endonuclease (EN) and reverse-transcriptase activities and a cysteine-rich domain that is also essential for retrotransposition [1,14]. In the cytoplasm, L1ORF2p proteins preferentially bind to the poly-A stretch of the LINE1 mRNA that they are encoded by and, together with the mRNA-binding L1ORF1p proteins, form ribonucleoprotein (RNP) complexes that enter the nucleus [1]. The EN domain of L1ORF2p nicks one strand of host genomic DNA at a consensus EN recognition sequence (usually “5′-TTTT/A-3′”), exposing the poly-T sequence that is annealed to the poly-A stretch of the substrate RNA and providing a free 3′-hydroxyl group that is used to initiate target-primed reverse transcription (TPRT) [1,15]. At the integration site, the break in the second strand of the genomic DNA usually occurs 7–20 nucleotides downstream of the initial single-strand nick, producing target-site duplications (TSDs) flanking LINE1-mediated retrotransposition events [1,15]. As a consequence of TPRT, about 95% of all LINE1 retrotransposition events are 5′-truncated, and all genomic 3′ junctions are characterized by a poly-A tail adjacent to an EN recognition site [1,15].

LINE1 proteins can also act in *trans* and retrotranspose RNA encoded by non-autonomous Alu retrotransposons and mRNA encoded by RNA-Pol II host genes, with about 10 to 100-fold and 3000 to 10,000-fold lower efficiency than LINE1 mRNA in cell culture assays, respectively [1,16,17,18]. Distinct mechanisms, including specific RNA-binding factors and template choice/switching, have been shown to be involved in the LINE1-mediated retrotransposition of cellular RNAs [18,19,20].

Polyadenylated viral mRNAs, which are usually highly amplified during virus infections, can be targets of LINE1-mediated retrotransposition. Non-retroviral RNA virus sequences have been discovered in the human genome, with characteristics, including a TSD and an integrated poly-A tract, that are indicative of a LINE1-mediated TPRT mechanism [21,22,23]. SARS-CoV-2 is a positive-strand RNA virus. In SARS-CoV-2 infected cells, viral replication and transcription produce large amounts of positive-strand genomic RNA and sub-genomic mRNAs (Appendix A, reviewed in references [24,25]). These positive-strand RNAs, which have the same 3′-end and are polyadenylated (Appendix A), can reach up to 80% of total polyA-RNAs in infected cells [26], suggesting that they could be substrates for LINE1-mediated retrotransposition in *trans*. We have provided experimental evidence that SARS-CoV-2 subgenomic mRNAs can be retrotransposed into the host cell genome through a LINE1-mediated TPRT mechanism in cell culture [27]. Such retrotransposed viral DNA could be related to some of the reports of prolonged or recurrent RT-PCR positivity in some “long COVID” patients [27]. Nanopore long-read WGS was used to detect retrotransposed SARS-CoV-2 cDNAs in LINE1-overexpressing cultured cells after viral infection, with signature TSDs, poly-A tracts, and genomic sequence motifs adjacent to the poly-A tails of SARS-CoV-2 sequences that are consistent with the consensus LINE1 EN recognition sequence [27]. For two of the integrated viral DNAs, we recovered cellular sequences flanking both sides of the integrated viral DNA, with a TSD and a short poly-A tract directly linked to a LINE1 endonuclease (EN) recognition sequence (Appendix A and reference [27]). One of the retrotransposed viral sequences was a full-length DNA copy of the nucleocapsid subgenomic mRNA (~1.7 kb, Appendix A and reference [27]). The parental mRNA has a viral leader sequence fused to the body of the nucleocapsid gene at the transcription regulatory sequence (TRS), generated by discontinuous transcription by the viral RNA polymerase (see architectures for SARS-CoV-2 mRNAs [24,25,28], Appendix A). The integrated viral cDNA detected by Nanopore sequencing matched exactly the sequence of the nucleocapsid subgenomic mRNA (Appendix A and reference [27]). The second retrotransposed sequence was a 5′-end truncated copy of viral RNA that included the 3′ poly-A end and reached the middle of the nucleocapsid gene (~0.55 kb, Appendix A and reference [27]), consistent with 5′-truncation being a common feature of LINE1-mediated retrotranspositions [1].

In cultured cells that did not overexpress LINE1, WGS was not sensitive enough to detect integrated viral DNA, as we previously found and others showed [29], and a target enrichment and sequencing method, TagMap [30,31,32,33], was used to detect retrotransposed viral DNAs. The retrotransposed sequences detected by TagMap had a 3′-end junction that was linked to cellular sequences with a poly-A tract, and the host DNA had LINE1 EN recognition sequences [27]. Because, in the absence of LINE1 overexpression, the retrotransposition of SARS-CoV-2 sequences was much rarer, our results were controversial [29,34,35,36,37]. 

This article has two goals: (1) To confirm our previous results and to systematically compare the relative sensitivities of whole-genome and enrichment sequencing for the detection of rare SARS-CoV-2 RNA retrotransposition events; (2) to investigate whether viral mRNA transfected into cells can also be retrotransposed into genomic DNA. Our results suggest that viral infection can lead to endogenous LINE1 de-repression and thus stimulate the retrotransposition of viral RNA. In contrast, we did not detect retrotranspositions following the transfection of viral RNA into cultured cells that did not overexpress LINE1.

## 2. Materials and Methods

### 2.1. Cell Culture, Transduction, and Transfection

The cell culture for HEK293T and Calu3 cells and LINE1 overexpression in HEK293T cells were described previously [27]. The human embryonic stem cell (hESC) line WA01/H1 was obtained from WiCell (Madison, WI, USA). hESCs were maintained feeder-free on Matrigel (Corning, Bedford, MA, USA; 354234) in StemFlex Medium (Thermo Fisher Scientific, Waltham, MA, USA; A3349401) and passaged in cell aggregates using Versene Solution (Thermo Fisher Scientific; 15040066) following the manufacturers’ instructions. 

Lentiviruses were produced using a plasmid carrying the human ACE2, pLENTI_hACE2_HygR, a gift from Raffaele De Francesco (Addgene plasmid # 155296; http://n2t.net/addgene:155296; RRID: Addgene 155296; accessed on 28 March 2022) [38], with third generation helper and transfer plasmids, following standard procedures. hESCs were transduced with the ACE2-expressing lentiviruses. ACE2 expression was confirmed 24 h after transduction by RT-qPCR and Western blot. 

Vascular smooth muscle cells were differentiated from the H1 hESC line (WiCell) as described previously [39]. Briefly, H1 cells were plated at 15,000 cells/cm^2^ on Matrigel-coated plates. The next day differentiation was started by the addition of 6uM Chir99021 (Cayman Chemical, Ann Arbor, MI, USA; Item No. 13122) and 25 ng/mL BMP4 (PeproTech, Cranbury, NJ, USA; 120-05ET) in Advanced DMEM/F12 (Gibco, Grand Island, NE, USA; 12634010). The medium was changed daily for 3 days. Afterward, the cells were treated daily with 10ng/mL PDGF-BB (PeproTech; 100-14B) and 2 ng/mL Activin A (PeproTech; 120-14E) in Advanced DMEM/F12 for 2 days. The cells were then passaged and maintained in Human Vascular Smooth Muscle Cell Basal Medium (Gibco; M231500) with added Smooth Muscle Growth Supplement (Gibco; S00725) on fibronectin (MilliporeSigma, Darmstadt, Germany; F0556)-coated plates. For reverse transcriptase inhibitor (RTi) treatment, 100 μM (final concentration) Azidothymidine (AZT) (Tocris Bioscience, Bristol, UK; 4150/50) and 10 μM (final concentration) Abacavir hemisulfate (ABC) (Tocris Bioscience; 4148/10) were added to cell culture.

The cell transfection of DNA, RNA or Poly(I:C) HMW (InvivoGen, San Diego, CA, USA; tlrl-pic) was carried out with Lipofectamine 3000 (Invitrogen, Waltham, MA, USA; L3000001 or L3000008) following the manufacturer’s protocol.

### 2.2. Virus Infection

SARS-CoV-2 infection for HEK293T and Calu3 cells has been described previously [27]. For viral infection in hESCs, the SARS-CoV-2 NeonGreen virus was obtained through BEI Resources, NIAID, NIH: SARS-Related Coronavirus 2, Isolate USA-WA1/2020, Recombinant Infectious Clone with Enhanced Green Fluorescent Protein (icSARS-CoV-2-eGFP), NR-54002. The viral stocks were prepared in Vero E6 cells (ATCC, Manassas, VA, USA; CRL-1586) cultured in Dulbecco’s modified Eagle’s medium (DMEM) supplemented with 2% fetal calf serum (FCS), penicillin (50 U/mL), and streptomycin (50 mg/mL). hESCs transduced with human ACE2 (hESCs-ACE2) were infected with the virus at a multiplicity of infection (MOI) of 1. Infected hESCs-ACE2 cells were harvested at 72 h post-infection for DNA isolation. All work with SARS-CoV-2 was performed in the biosafety level 3 (BSL3) laboratory at the Ragon Institute (Cambridge, Massachusetts) following approved standard operating procedures (SOPs).

### 2.3. Nucleic Acid Extraction

Cellular DNA extraction was performed as previously described [27] or using the Wizard HMW DNA Extraction Kit (Promega, Madison, WI, USA; A2920) following the manufacturer’s protocol with modifications that RNase and Proteinase treatments were each extended to one hour. Cellular RNA was extracted with RNeasy Plus Micro or Mini Kit (Qiagen, Hilden, Germany; 74034 or 74134) following the manufacturer’s protocols.

### 2.4. WGS and Analysis

The Nanopore WGS of DNA from hESCs after ACE2 transduction and SARS-CoV-2 infection was performed as previously described [27], except that the SQK-LSK110 kit (Oxford Nanopore Technologies) was used for the construction of the sequencing library instead of the previously used SQK-LSK109 kit [27]. Sequencing read alignment to the human and SARS-CoV-2 genomes was performed as previously described [27]. The number of mapped bases was obtained by running samtools stats (version 1.11, http://www.htslib.org/doc/samtools-stats.html) (accessed on 2 November 2022) [40] on the aligned SAM file. Sequencing genome coverage was calculated by dividing the mapped base number by the human genome size (3,200,000,000 bp). 

Illumina WGS of DNA from Calu3 cells after SARS-CoV-2 infection, as well as sequencing read alignment, was performed as previously described [27]. Sequencing genome coverage was calculated by multiplying the mapped read–pair number (obtained from the read alignment report) with the read–pair length (2 × 150 bp) and then dividing the result by the human genome size (3,200,000,000 bp).

### 2.5. TagMap and Analysis

TagMap experiments were performed as previously described [27]. The alignment for raw sequencing reads or duplicate-removed reads [by dedup_hash (https://github.com/mvdbeek/dedup_hash), accessed on 19 March 2021] and integration analysis were performed as previously described [27]. For the same datasets, we also compared different parameters to call chimeric reads with STAR [41] (version 2.7.1a): \–chimSegmentMin 40 \–chimJunctionOverhangMin 40; or \–chimSegmentMin 125 \–chimJunctionOverhangMin 125. The read alignment illustrations in the figures were generated using the UCSC genome browser and Adobe Illustrator 2022 (Adobe, San Jose, CA, USA). 

For TagMap analysis in LINE1-overexpressing 293T cells, the genomic distances (in bps) between the closest LINE1 EN recognition sequence (a “TTTT/N” sequence) and the mapped human sequence end (Appendix A) were measured manually in the UCSC genome browser. To calculate the expected probability of seeing a LINE1 EN recognition sequence (“TTTT/N”) within a certain distance (n) from a mapped human read end (fixed point) (Appendix A), the following formula was used: 1 − (1 − 1/256) ^ n. In this formula, the probability of not randomly generating a given k-mer (k = 4 for “TTTT”) is p = 1 − (1/4) ^ 4 = 1 − 1/256. Then, the chance of not seeing it after giving n tries (within a distance n bases from a fixed point) is p ^ n. Therefore, the expected probability of seeing this k-mer (“TTTT”) at a distance from a fixed point is 1 − p ^ n.

To display the sequence patterns of the 3′-end virus–host junctions, sequence logos were generated using the program WebLogo [42] (https://weblogo.berkeley.edu/logo.cgi) (accessed on 18 February 2023), as described in a previous publication [43].

### 2.6. Digital PCR (dPCR)

dPCR was performed using the Qiagen QIAcuity Digital PCR system (Instrument: QIAcuity-00412; Software: QIAcuity Software Suite 2.1.7.182) and the QIAcuity EG PCR Kit (Qiagen; 250111) following the manufacturer’s protocols. The following dPCR parameters were used: Step 1, QIAGEN Standard Priming Profile; Step 2, cycling profile with 1 cycle of 95 °C 2 min, 40 cycles of (95 °C 15s, 60 °C 15s, 72 °C 15 s), and 1 cycle of 35 °C 5 min; and Step 3, imaging with the “Green” channel in the system, with an exposure duration of 200 ms and imaging gain of 6. QIAcuity Nanoplate 8.5k 96-well (Qiagen; 250021) was used for LINE1-overexpressing 293T cell DNA. QIAcuity Nanoplate 26k 24-well (Qiagen; 250001) was used for hESC-ACE2 and Calu3 cell DNA. The selected SARS-CoV-2 primer pair (Forward: ACGCGGAGTACGATCGAG; Reverse: TATTAAAATCACATGGGGATAGCAC) targets a 113 bp sequence near the 3′-end (poly-A) of the SARS-CoV-2 (sub)genomic RNA with validated specificity. A primer pair targeting a 78-bp sequence of the human TUBB gene (Forward: TCCCTAAGCCTCCAGAAACG; Reverse: CCAGAGTCAGGGGTGTTCAT) was used as an internal control with validated specificity. Both no-template control and DNA from mock-infected or non-infected control cells were used as negative controls, showing no specific amplification of viral cDNA.

### 2.7. RNA-Seq and Analysis

Poly-A RNA-seq and data analysis were performed using the same methods as previously described [27].

### 2.8. RNA In Vitro Transcription (IVT)

For the IVT of the nucleocapsid mRNA (with the viral leader sequence), the pUC57-2019-ncov plasmid [44], a kind gift from Christine A. Roden from the Amy S. Gladfelter laboratory (University of North Carolina at Chapel Hill), was used as the template DNA. In this plasmid, a T7 promoter is followed by the full-length nucleocapsid subgenomic mRNA sequence, including the viral leader sequence, nucleocapsid ORF, 3′-UTR, and a short 25-nt poly-A sequence, which is then followed by a SalI (NEB, Ipswich, MA, USA; R0138L) restriction site. This plasmid was linearized and fragmented by SalI restriction digestion (there is a second SalI restriction site upstream of the T7 promoter), which was confirmed by agarose gel electrophoresis. To generate the IVT template for the nucleocapsid mRNA without the viral leader sequence, the nucleocapsid subsequence starting from the start (ATG) of the nucleocapsid ORF to the ending 25-nt poly-A was cloned into a pGEM-7Zf(+) vector (Promega; P2251) by XbaI (NEB; R0145L) and BamHI (NEB; R3136L) restriction cloning. This plasmid was linearized by BamHI restriction digestion, which was confirmed by agarose gel electrophoresis. 

The linearized/fragmented plasmid DNA to be used as an IVT template was concentrated by ethanol precipitation. IVT for capped RNA was performed using the mMESSAGE mMACHINE™ T7 ULTRA Transcription Kit (Invitrogen; AM1345) following the manufacturer’s protocol. The IVT for uncapped RNA was conducted with the same kit/method, except for the replacement of 2× NTP/ARCA with individual NTPs from the MEGAscript™ T7 Transcription Kit (Invitrogen; AM1333). After IVT, the RNA sample was treated with DNase to remove the template DNA and was then polyadenylated, following protocols from the mMESSAGE mMACHINE™ T7 ULTRA Transcription Kit (Invitrogen; AM1345). The size and polyadenylation of the desired RNA products were confirmed by gel electrophoresis. Finally, the RNA was column-purified using an RNeasy Plus Mini Kit (Qiagen; 74134) and was eluted with water.

### 2.9. Reverse Transcription-Quantitative Polymerase Chain Reaction (RT-qPCR)

To quantify LINE1 mRNA, RT-qPCR on purified cellular poly-A RNA was performed following the protocol in a previous publication [2]. The total cellular RNA was extracted using an RNeasy Plus Mini Kit (Qiagen; 74134) with extensive RNase-free DNase (Qiagen; 79254) digestion (37 °C 30 min). Poly-A RNA was isolated from the total RNA using the NEBNext Poly(A) mRNA Magnetic Isolation Module (NEB; E7490L or E7490S) and then quantified by a Qubit 3.0 Fluorometer (ThermoFisher) with RNA high-sensitivity mode. The isolated poly-A RNA was reverse-transcribed by qScript cDNA SuperMix (QuantaBio; Beverly, MA, USA; 95048-500), following the manufacturer’s protocol. The removal of the cellular DNA from the isolated poly-A RNA was confirmed by qPCR controls that omitted the RT enzyme. qPCR was conducted using PowerUp SYBR Green Master Mix (Applied Biosystems, Waltham, MA, USA; A25742) in a QuantStudio 6 system (Applied Biosystems). GAPDH was used as an internal control for normalization. LINE1 and GAPDH primer sequences were following the previous publication [2]: LINE1 5-UTR: TAAACAAAGCGGCCGGGAA and AGAGGTGGAGCCTACAGAGG; LINE1 ORF1: ACCTGAAAGTGACGGGGAGA and CCTGCCTTGCTAGATTGGGG; LINE1 ORF2: CAAACACCGCATATTCTCACTCA and CTTCCTGTGTCCATGTGATCTCA; GAPDH (intron spanning): TTGAGGTCAATGAAGGGGTC and GAAGGTGAAGGTCGGAGTCA. Three independent experiments (biological replicates) were performed for each treatment versus the control comparison. The mean cycle threshold (Ct) value of qPCR triplicates from each experiment (biological replicate), normalized to the internal control gene GAPDH, was used. Statistical analyses were undertaken to test if there were significant up-regulations of LINE1 mRNA in treated (infected, transfected or hydrogen peroxide treated) versus the control cells, using one-tailed, unpaired *t*-tests and assuming equal standard deviation. 

To quantify and compare the amount of 5′-capped versus uncapped nucleocapsid mRNA in the cells after transfection, the total 293T cellular RNA was extracted 1-day post-transfection by RNeasy Plus Mini Kit (Qiagen; 74134) with RNase-free DNase (Qiagen; 79254) digestion (37 °C 15 min). RNA was reverse-transcribed by qScript cDNA SuperMix (QuantaBio; 95048-500), following the manufacturer’s protocol. qPCR was conducted using PowerUp SYBR Green Master Mix (Applied Biosystems; A25742) in a QuantStudio 6 system (Applied Biosystems). Three independent experiments (biological replicates) were performed. The mean cycle threshold (Ct) value of the qPCR triplicates from each experiment (biological replicate) was normalized using measurements of RNA from the host gene GAPDH. Statistical analyses were undertaken to test if there were significant differences in the amount of the transfected capped versus uncapped nucleocapsid mRNA, using two-tailed, unpaired *t*-tests and assuming equal standard deviation. The following nucleocapsid primers were used, with validated specificity: amplicon 1: GGGAGCCTTGAATACACCAAAA and TGTAGCACGATTGCAGCATTG; amplicon 2: GGGGAACTTCTCCTGCTAGAAT and CAGACATTTTGCTCTCAAGCTG; amplicon 3: ACGCGGAGTACGATCGAG and TATTAAAATCACATGGGGATAGCAC. RNA from mock-transfected cells and no template controls (water) were performed as negative controls, showing no specific amplification of the nucleocapsid sequences.

All RT-qPCR data plotting and statistics were performed using Prism 9 software (version 9.4.1, GraphPad Software, LLC., San Diego, CA, USA).

### 2.10. Cell Immunofluorescence Staining

For immunofluorescence staining, the cells were grown on 12 mm round coverslips and fixed with 1.6% (*w*/*v*) paraformaldehyde/CMF-PBS at room temperature (RT) for 15 min. The cells were permeabilized with 0.5% (*v*/*v*) Triton X-100/PBS (PBST), washed with 0.1% PBST three times, blocked with 4% (*w*/*v*) BSA/CMF-PBS, and then incubated with primary antibodies. The cells were then washed with 0.1% PBST three times, incubated with secondary antibodies, and washed with 0.1% PBST three times. The primary antibodies used in this study are: anti-LINE1ORF1p mouse monoclonal antibody (clone 4H1, MilliporeSigma; MABC1152; 1:400 dilution); anti-G3BP1 rabbit polyclonal antibody (ThermoFisher; 13057-2-AP; 1:600 dilution); anti-SARS-CoV-2 Nucleocapsid rabbit polyclonal antibody (GeneTex, Irvine, CA, USA; GTX135357; 1:1000 or 1:600 dilution). For LINE1ORF1p staining, donkey anti-mouse IgG Alexa Fluor 594 secondary antibody (Invitrogen; A-21203; 1:600 dilution) was used. For Nucleocapsid immunostaining, donkey anti-rabbit IgG Alexa Fluor 488 secondary antibody (Invitrogen; A-21206; 1:1000 dilution) was used. For LINE1ORF1p and G3BP1 co-staining, donkey anti-mouse IgG Alexa Fluor 488 secondary antibody (Invitrogen; A-21202; 1:1000 dilution) combined with donkey anti-rabbit IgG Alexa Fluor 647 secondary antibody (Invitrogen; A-31573; 1:1000 dilution) were used. The cells on the coverslips were mounted with VECTASHIELD HardSet Antifade Mounting Medium with DAPI (Vector Laboratories, Newark, NJ, USA; H-1500-10). The 3D optical sections were acquired with 0.2-μm z-steps using a DeltaVision Elite Imaging System microscope system with a 100× oil objective (NA 1.4) and a pco.edge 5.5 camera and DeltaVision SoftWoRx software (version 7.0.0, GE Healthcare, Chicago, IL, USA). Image deconvolution was performed using SoftWoRx. All of the figure panel images were prepared using FIJI software (ImageJ, version 2.1.0/1.53c, NIH) and Adobe Illustrator 2022 (Adobe), showing deconvolved single z-slices.

## 3. Results

### 3.1. WGS Can Be Used to Detect Reverse-Transcribed Viral cDNA in SARS-CoV-2 Infected Cells but the Sensitivity of Detection Is Limited by the Depth of Sequencing

To increase the probability that retrotransposed SARS-CoV-2 sequences would be generated, we previously analyzed infected HEK293T (293T) cells that overexpressed a functional LINE1 element [27]. Nanopore WGS identified SARS-CoV-2 cDNA sequences retrotransposed in the genomic DNA of the infected cells with LINE1 overexpression [27]. However, there needs to be a more accurate measurement of the numbers of reverse-transcribed viral cDNAs in cells that do, and do not, overexpress LINE1. Digital PCR (dPCR) is a highly precise and sensitive method to quantify nucleic acids, dividing the sample into thousands of partitions using fluidics technologies [45,46]. We used dPCR to estimate the frequency of SARS-CoV-2 cDNA in infected cells, using primers targeting a sequence within a 500nt segment from the 3′-end of the SARS-CoV-2 genomic/subgenomic RNA (see Materials and Methods). Using the same DNA from infected LINE1-overexpressing 293T cells that was used in the previous publication [27], we found that there were between 10,000 and 20,000 copies per 1000 cells of SARS-CoV-2 cDNAs derived from the 3′-end of the viral RNA sequence (Table 1). This analysis gives a direct measure of the number of reverse-transcribed viral cDNAs in the LINE1-overexpressing cells that can be compared to the results obtained by sequencing methods (WGS and TagMap).

We previously reported a total of 63 instances of retrotransposed viral cDNA in the LINE1-overexpressing 293T cells after infection, based on the detection of host–virus DNA junctions by Nanopore WGS [27]. For 32 retrotranspositions in which the 3′-end integration junction was detected, we found a poly-A tract ranging from 2 to 65 bp that was directly linked to a LINE1 EN recognition sequence (e.g., 5′-TTTT/A-3′), supporting a LINE1-mediated retrotransposition mechanism [27]. These results suggest that at least 32 retrotransposed viral cDNAs were present based on a sequencing depth covering ~18 haploid genomes (~3600 copies per 1000 cells, Table 1).

To determine the sensitivity of the Nanopore WGS method, we counted the total number of viral cDNAs in infected LINE1-overexpressing 293T cells, using data from reference [27], which contain viral sequences that can be mapped to the 3′-end 500nt of the SARS-CoV-2 genomic/subgenomic RNA sequence (upstream of the poly-A sequence). We identified 233 unique reads from the PCR-independent whole-genome sequencing to a depth of ~18 haploid genome. This result suggested that there were 233 viral cDNAs, reverse transcribed from the 3′-end of SARS-CoV-2 RNA, detected in ~9 infected cell DNA equivalents (a frequency of ~26,000 copies per 1000 cells, Table 1), which supports the dPCR data. This number represents the total number of SARS-CoV-2 cDNA copies and thus is the sum of retrotransposed cDNA copies (detected with flanking human sequences) generated through a LINE1 TPRT mechanism [27] and reverse transcribed but potentially unintegrated cDNA copies (detected as “viral-only” sequences). It has been reported by others that LINE1 reverse transcriptase can generate extrachromosomal, cytoplasmic LINE1 or Alu cDNAs [2,3,4,47,48]. The average length of “viral-only” reads was 1.4 kb, while the average read length in the entire dataset (all human and viral reads) was 5.3 kb, consistent with the interpretation that most of the “viral-only” reads were unintegrated extrachromosomal DNA. Our results suggest that 10 to 20% of the total viral cDNA copies are retrotransposed (Table 1).

Our results suggest that Nanopore has good sensitivity for the detection of viral cDNAs. However, the sequencing depth of the WGS-based method limits the number of cells that can be analyzed. Although 1–2 μg of input DNA (corresponding to about 200,000 cell genomes) was used to construct the Nanopore sequencing library [27], sequences corresponding to only ~18 complete human haploid genomes were recovered based on a WGS depth of 18-fold. Thus, WGS is unlikely to detect reverse-transcribed viral sequences if the number of sequenced cells is less than the number of cells that carry one viral cDNA. In fact, we failed to detect viral cDNAs in SARS-CoV-2 infected cells that did not overexpress LINE1 by WGS, as summarized in Table 1 and consistent with previous publications [27,29]. No viral cDNA was detected using WGS on DNA from infected Calu3 cells or human embryonic stem cells (hESCs) that had been transduced with the ACE2 receptor to enable efficient viral infection. However, dPCR detected about four copies of viral cDNA per 1000 hESCs and about 14–21 copies per 1000 Calu3 cells (Table 1). These estimates suggest that approximately one copy of viral cDNA was present in 50–250 cells in the two cell lines when LINE1 was not overexpressed. This also suggests that the overexpression of LINE1 increases the reverse transcription of viral RNA by about 1000×. WGS coverage needs to reach at least 150–500× of human genome equivalents to detect a single viral cDNA sequence (Calu3 is near-triploid, and hESC is diploid) to detect a single viral cDNA sequence. To detect retrotransposed viral cDNA in cells that did not overexpress LINE1, we used TagMap [30,31,32,33], which can enrich for host–virus junctions and interrogate 1000–2000-fold more cells than WGS (Table 1, discussed below).

### 3.2. TagMap Can Detect LINE1-Mediated Retrotransposition of SARS-CoV-2 RNA by Enriching for Host–Virus DNA Junctions

To compare the ability of Nanopore WGS and TagMap [30,31,32,33] to detect viral RNA retrotranspositions, we applied TagMap to the same DNA samples extracted from infected LINE1-overexpressing 293T cells that had been used for Nanopore WGS in our previous publication [27]. In two independent virus infection experiments (biological replicates), the TagMap method identified 33 and 54 chimeric sequences consisting of a LINE1 endonuclease (EN) recognition sequence in the host sequences that was directly linked to a viral poly-A sequence (poly-A tract; Figure 1 and Appendix A), which are characteristic of LINE1-mediated retrotransposition events. The sequences near the 3′-junctions identified by TagMap have been summarized using Sequence Logos (Appendix A) and are consistent with the sequences near the 3′-junctions identified by Nanopore WGS in our previously published data [27] (Appendix A). We also detected chimeric DNA sequences in which one end of the DNA mapped to human sequences and the other end mapped to the 3′-end (containing the poly-A sequence) of the viral RNAs (645 and 1056 chimeric sequences were detected in two independent infection experiments, Appendix A). Although a poly-A tract was present in these mapped sequences, the exact sites of the retrotranspositions (including the LINE1 EN recognition sequences) were not directly identified due to the read length limitation of Illumina sequencing (Appendix A). However, the chance of there being a LINE1 EN recognition sequence (e.g., TTTT/A) within 500 base-pairs from the mapped human sequence is significantly higher than the probability that a random “TTTT” motif sequence would be present (Appendix A), consistent with these chimeric DNA sequences most likely having been generated by the LINE1-mediated TPRT mechanism. In total, we detected 678 and 1110 retrotransposition events by TagMap in LINE1-overexpressing 293T cells from two independent SARS-CoV-2 infection experiments derived from about 4000 cells (estimated from the amount of input DNA in each replicate, a frequency of ~170–280 copies per 1000 cells, Table 1). Based on the number of cells analyzed, the Nanopore and dPCR methods imply that there were about 20,000 total cDNA copies derived from viral 3′-end RNA per 1000 cells and about 3600 copies (~10–20%) of retrotransposed cDNA with a sequence from the 3′ end of the viral RNA per 1000 cells (Table 1). Because TagMap specifically detects the 3′-end viral–host junctions, these results suggest that TagMap can recover 5–10% of retrotransposed viral cDNAs (Table 1). However, it is likely that (1) there were DNA losses during the TagMap procedure, and thus the input cell numbers, based on the amount of DNA that was used for PCR enrichment, may be an overestimate; and (2) the selective amplification of the viral–host junctions was probably less than 100% because one primer of the enrichment primer-pair targeted an adapter that was randomly inserted into the host genomic DNA by Tn5 tagmentation, and the primers would not cover all possible retrotransposition junctions.

In cells that did not overexpress LINE1, consistent with our previous publication [27], TagMap detected ~0.05 copies of TPRT-retrotransposed viral cDNA per 1000 hESC-ACE2 and ~0.08–0.17 copies of TPRT-retrotransposed viral cDNA per 1000 Calu3 cells (Table 1). Assuming that the efficiency (~5–10%) of recovering the retrotransposed viral cDNAs with sequences from the 3′ end of viral RNA by TagMap is similar in cells with or without LINE1 overexpression, the results in Table 1 imply that about one copy of TPRT-retrotransposed viral cDNA was carried in 1000 cells. If we make the assumption that, similar to what was seen in the cells that overexpressed LINE1, about 10–20% of the viral 3′ end sequences were retrotransposed (Table 1), the TagMap data are also good matches to the dPCR data (about ten copies of viral cDNAs carried in 1000 cells). Thus, our results suggest that an integration enrichment method such as TagMap, but not WGS, can be used to detect integrations in a small fraction of the cell population.

### 3.3. Viral RNA Transfection Alone Did Not Show Retrotranspositions in Cultured Cells by TagMap

To ask what viral–host interactions might affect viral RNA retrotransposition, we established a viral RNA transfection system and tested what condition(s) could affect viral RNA retrotransposition. Because nucleocapsid has been reported to be the most abundant viral mRNA in infected cells [28], and because a full-length retrotransposed cDNA copy of this RNA species has been identified (Appendix A and reference [27]), we synthesized SARS-CoV-2 nucleocapsid subgenomic mRNA (Appendix A and Figure 2A) by in vitro transcription (IVT). We also produced nucleocapsid mRNA lacking the viral leader sequence (Figure 2B) to ask whether this viral-specific non-coding sequence could affect retrotransposition.

The nucleocapsid protein that is translated from the nucleocapsid mRNA is an RNA-binding protein that packages the viral genome and can stabilize viral RNAs and may also be involved in interactions with host cell proteins [25,44]. To control the translation efficiency of nucleocapsid proteins from transfected mRNAs, we produced either 5′-capped or uncapped versions of the nucleocapsid mRNAs. After transfection using a commercial lipofectamine reagent, the capped mRNAs (with or without the 5′ leader sequence) expressed nucleocapsid protein that could be detected with immunofluorescence staining, while uncapped mRNAs did not express detectable levels of nucleocapsid proteins, as expected (Appendix A).

We followed the same TagMap protocol to detect 3′-end viral–host DNA junctions (Figure 1 and Appendix A) using the DNA extracted from 293T cells harvested 3 or 6 days after the nucleocapsid mRNA was transfected. In three independent transfection experiments, we failed to detect any retrotransposition events in cells transfected with 5′-capped or uncapped nucleocapsid mRNA (Figure 2A,C). As a positive control, we detected retrotranspositions of transfected nucleocapsid mRNA in 293T cells that overexpressed LINE1 by co-transfection of a LINE1 expression plasmid (Figure 2A,B). There was no significant difference in the efficiency of retrotransposition of nucleocapsid mRNA with (Figure 2A) or without (Figure 2B) the 5′ leader sequence when LINE1 was overexpressed. In a total of six experiments, we found that transfection of 5′-capped nucleocapsid mRNA (which can express the nucleocapsid protein) in LINE1-overexpressing cells resulted in ~5–10 times more retrotransposition events than transfection of uncapped nucleocapsid mRNA (Figure 2A,B), suggesting that the nucleocapsid protein may enhance the retrotransposition of viral mRNA. There was no significant difference in the amount of transfected RNA on 1-day post-transfection in cells transfected with 5′-capped and non-capped nucleocapsid mRNA (Appendix A), suggesting the difference in TagMap detected retrotransposition events was not related to a change in RNA stability that was a result of 5′-cap.

Similarly, in a different cell type, vascular smooth muscle cells (non-contractile, synthetic phenotype) differentiated from the hESC line H1, we detected no retrotransposition events using TagMap in cells harvested 3 days after NC mRNA (5′-capped) transfection (Appendix A). In the positive controls, eight retrotransposition events were detected by TagMap when the LINE1 expression plasmid was co-transfected, and three retrotransposition events were detected when the co-transfected cells were treated with RT inhibitors AZT and ABC (Appendix A). These RT inhibitors have been reported to be able to block LINE1 reverse transcriptase activities and LINE1 retrotranspositions [49]. Because the number of retrotransposition events that were detected in the RT inhibitor experiment was small, the significance of these data is unclear. However, in all the experiments that we performed in which NC mRNA was transfected into cells that did not overexpress LINE1, there was no detectable NC RNA retrotransposition.

We found that, in the highly infectable cell line Calu3, the level of SARS-CoV-2 RNA can be as high as ~80% of the total cellular polyA-RNA after infection (Figure 2D), possibly as a result of viral mRNA amplification and host mRNA degradation, which is consistent with previously published data [26]. In comparison, in 293T cells, a highly transfectable cell line, the abundance of transfected nucleocapsid mRNAs (using lipofectamine) was less than 2% of the total cellular polyA-RNA pool at 6 h post-transfection (a timepoint with the highest amount of delivered RNA) (Figure 2D). It is likely that the large fraction of SARS-CoV-2 polyadenylated mRNA in infected cells, relative to transfected cells, increases the likelihood that a viral RNA can interact with LINE1 proteins in *trans* and be retrotransposed. This difference in viral RNA level possibly explains that in LINE1-overexpressing cells, retrotransposed copies in the 5′-capped nucleocapsid mRNA transfected cells were ~10 times lower than that in virus-infected cells (Figure 2C).

### 3.4. SARS-CoV-2 Infection, but Not Nucleocapsid mRNA Transfection, Can Induce Endogenous LINE1 Expression in 293T Cells

We tested whether SARS-CoV-2 infection or related cellular stresses can induce endogenous LINE1 expression. LINE1 mRNA was measured by RT-qPCR using purified cellular polyadenylated RNAs, following previously published Human/Primate-specific LINE1 [LINE1HS/PA (2-6)] primers and protocol [2], targeting the 5′UTR, ORF1 or ORF2 regions of endogenous LINE1. We found that SARS-CoV-2 infection led to a modest increase in LINE1 mRNA (about 2-fold) in Calu3 cells (Figure 3A, *p* = 0.098 for L1-5′UTR, *p* = 0.021 for L1-ORF1, *p* = 0.303 for L1-ORF2). This result is consistent with previously published results showing that L1ORF1p protein was induced ~1.5-fold in tissues of COVID-19 patients [7] and that LINE1 retrotransposition or LINE1-mediated *trans*-mobilization of Alu and SVA elements were increased in SARS-CoV-2 infected 293T cells as compared to mock control [29].

In SARS-CoV-2-infected cells, both single-stranded viral genomic/subgenomic RNAs and double-stranded RNA intermediates are produced [24,25]. We found that the transfection of in vitro transcribed SARS-CoV-2 nucleocapsid mRNA (Figure 2A) had a negligible impact on the level of endogenous LINE1 mRNA in 293T cells 24 h post-transfection (Figure 3B). However, the transfection of a synthetic analog of double-stranded RNA [Poly(I:C), high molecular weight, 1.5–8 kb] led to a modest 2–4-fold increase in LINE1 mRNA in 293T cells, 24 h post-transfection (Figure 3C, *p* = 0.143 for L1-5′UTR, *p* = 0.121 for L1-ORF1, *p* = 0.020 for L1-ORF2). The expressed LINE1ORF1p proteins formed large aggregates in the cytoplasm of Poly(I:C) transfected cells (Figure 3D). These aggregated LINE1ORF1p proteins were likely components of cell stress granules (SG), as evidenced by the co-localization of the SG component protein G3BP1 (Figure 3D). SG formation is an anti-viral defense mechanism [50] that has also been proposed to protect cells from potentially mutagenic retrotransposition events by sequestering LINE1 RNPs [51,52]. It has been reported that HCV infection can lead to endogenous LINE1 expression, but LINE1 retrotransposition is inhibited due to SG formation [11]. Many viruses have evolved diverse mechanisms to interfere with or prevent the formation and/or function of SG [50], which may inadvertently enhance retrotransposition. In SARS-CoV-2 infection, multiple mechanisms have been reported to inhibit SG formation and/or promote SG clearance [53,54,55,56,57,58]. One of the proposed mechanisms is that the SARS-CoV-2 nucleocapsid protein binds and sequesters the SG component G3BP1 [54,55].

SARS-CoV-2 infection can lead to the production of reactive oxygen species (ROS) [59,60,61], generating oxidative stress in cells. To mimic this stress, we treated cultured 293T cells with hydrogen peroxide and found a modest increase (~2–4-fold) of LINE1 mRNA (Appendix A, *p* = 0.112 for L1-5′UTR, *p* = 0.154 for L1-ORF1, *p* = 0.087 for L1-ORF2). Unlike Poly(I:C) transfected cells, the expressed LINE1ORF1p proteins were distributed in the cytoplasm of the hydrogen-peroxide-treated cells and did not form large aggregates (Appendix A).

## 4. Discussion

In this study, we compared WGS and TagMap as methods to detect retrotransposed SARS-CoV-2 sequences. Nanopore WGS is a sensitive way to detect retrotransposed viral cDNAs if they are carried in the small number of cells that are analyzed. However, because the number of genome equivalents that can be sequenced using Nanopore technology is limited, this method will not be able to detect retrotransposition events that have occurred in only a small fraction of the cells. In contrast, although the TagMap method is about one order of magnitude less efficient in recovering retrotransposed DNA that has sequences from the 3′ end of the viral RNA when compared to Nanopore WGS, it can be used to analyze the DNA from a much larger number of cells (up to ~20,000 cells), which makes it a more sensitive way to detect rare retrotransposition events. Thus, the previously published negative result that Nanopore WGS did not detect retrotransposed viral DNA in cells that did not overexpress LINE1 [29] is fully consistent with our previous and current results. Because LINE1 overexpression causes a large increase in the number of reverse-transcribed viral sequences (approximately 1000×), a failure to detect retrotransposed viral DNA by Nanopore sequencing in cells that do not overexpress LINE1 does not contradict the positive results we obtained using Nanopore WGS (or TagMap) in cells that overexpress LINE1. This result also suggests that anything that increases the level of LINE1 expression in cells is also likely to cause a general increase in the level of retrotransposition, including the retrotransposition of viral RNA in infected cells.

Nanopore WGS can sensitively detect reverse-transcribed viral cDNA if the copy number is high enough to be detected in a small number of cells. In 293T cells that overexpressed LINE1, ~70–80% of viral cDNAs recovered by Nanopore sequencing consisted of only viral sequences. The viral-only sequences were significantly shorter than the average Nanopore reads. It has been reported that LINE1 cDNAs can accumulate in the cytoplasm of senescent cells by unknown mechanism(s) [2,3,4]. It was shown recently that Alu cDNAs could be formed in the cytoplasm of RPE cells by LINE1 reverse-transcriptase, with a demonstrated mechanism [47,48]. Cytoplasmic cDNAs can trigger inflammation [2,3,4,48], which could be another source of cellular stress in infected cells and lead to LINE1 de-repression in turn. These reverse-transcribed cytoplasmic viral cDNAs could also lead to genomic integrations, as has been reported for cDNA integration of an exogenous RNA virus recombined with an endogenous retrotransposon [62].

It has been established that cellular stress, for example, aging or viral infection, can induce LINE1 expression [2,3,4,7,8,9,10,11]. Upon SARS-CoV-2 infection, double-stranded RNA intermediates are generated during viral replication/transcription [24,25], and reactive oxygen species (ROS) can be produced in patient tissues [59,60,61], which could potentially induce endogenous LINE1 expression based on our in vitro experiments. A recent publication suggested that SARS-CoV-2 infection can lead to disruption of heterochromatin maintenance and induce LINE1 expression in the tissues of patients [7]. Consistent with this, RNA-seq analyses have shown transcriptional upregulation of LINE1 elements in tissues from SARS-CoV-2 infected patients and cultured cells [8,63]. Regarding LINE1 retrotransposition, it has been reported that LINE1 *cis-*retrotransposition or *trans*-mobilization of Alu and SVA elements were increased upon SARS-CoV-2 infection in cultured 293T cells, analyzed by Nanopore WGS [29]. However, others have reported that, in cultured 293T cells, LINE1 *cis-*retrotransposition activity was inhibited by several SARS-CoV-2 nonstructural proteins, using transfection-based in vitro LINE1 retrotransposition reporter assays [64], suggesting that there are complex viral–host interactions.

Although the LINE1-mediated retrotransposition of cellular mRNAs has been reported to be less frequent than LINE1/Alu retrotransposition [16,17,18], the high level of viral poly-A RNAs in infected cells could increase the probability that viral RNAs will interact successfully with LINE1 proteins. We found that the presence of viral nucleocapsid protein increased the frequency of viral RNA retrotransposition. Additionally, stress granule (SG) formation, a host anti-viral [50] and anti-retrotransposition [51,52] defense mechanism, is inhibited by SARS-CoV-2 infection through multiple mechanisms [53,54,55,56,57,58]. It has been proposed that the nucleocapsid protein can inhibit SG by sequestering G3BP1 [54,55], an essential component of the SG, which could facilitate the retrotransposition of viral RNA by LINE1 in infected cells (Figure 3E).

An important question is whether vaccine mRNA can be subject to LINE1-mediated retrotransposition. We found that the transfection of SARS-CoV-2 nucleocapsid mRNA into cultured 293T cells with a commercial lipofectamine reagent did not lead to a detectable level of retrotransposition. As a positive control, we detected retrotransposed transfected viral nucleocapsid mRNA in cells that overexpressed LINE1 (which, as noted above, should increase retrotransposition by about 1000×), although the retrotransposed copies were ~10 times lower than that in virus-infected cells. There are several possible explanations for the differences in the levels of retrotransposition in infected and transfected cells: (i) The relative abundance of viral RNA is almost two orders of magnitude higher in infected cells than in transfected cells, which would increase the probability of association with LINE1 proteins; (ii) virus infection, but not viral mRNA transfection, can induce endogenous LINE1 expression; (iii) multiple factors during SARS-CoV-2 infection can inhibit the antiviral/anti-retrotransposition function of stress granules [53,54,55,56,57,58], which could increase retrotransposition (Figure 3E).

Our data also suggest that dsRNA (which is formed during viral infection) could induce endogenous LINE1 expression, and dsRNA is known to induce innate immune responses in cells infected with RNA viruses [65,66]. This result also suggests that therapeutic mRNAs should be designed to avoid unwanted dsRNA (secondary) structures. RNA base modifications and other optimizations can help to dampen the host’s response to dsRNA [67].

A limitation of this study is that the retrotransposition experiments were carried out in established cell lines or cultured cells differentiated from an hESC line. It is important to emphasize that the process of transfecting viral RNA into cultured cells in our experiments differs from vaccine RNA delivery (lipofection versus lipid nanoparticle-mediated RNA delivery) and that the RNA sequence we used differs from the vaccine RNA sequence (Nucleocapsid versus Spike) [68,69]. More studies are needed using more physiological conditions or animal models. Furthermore, it remains to be elucidated whether other viral or host factors involved in the interactions of viral RNA with host retrotransposons affect the propensity of viral RNA to be retrotransposed.

## Figures and Tables

**Figure 1 viruses-15-00629-f001:**
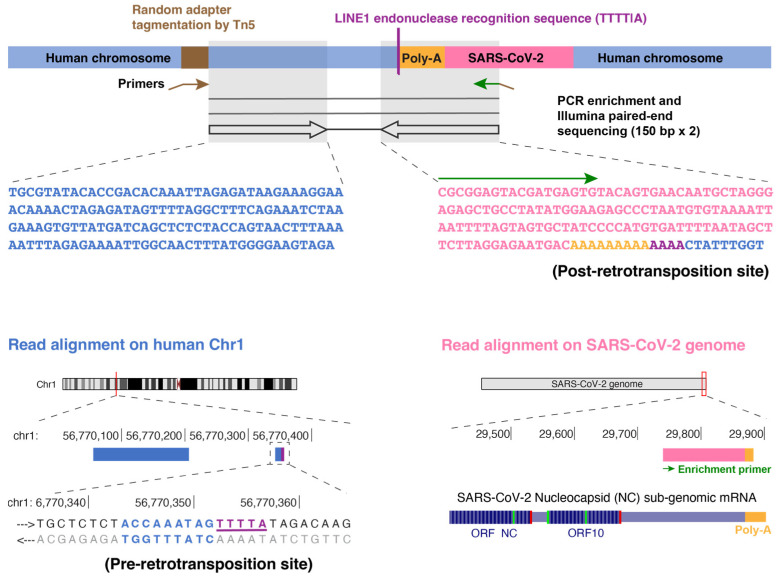
TagMap method for the enrichment and sequencing of a junction of retrotransposed viral cDNA sequence with its flanking human chromosome sequence. Schematic and an example sequencing read-pair showing the enrichment of one junction of retrotransposed viral cDNA. This method is based on random Tn5 tagmentation on cellular genomic DNA. A primer targeting the inserted adapter sequence (brown arrow) and a primer targeting a SARS-CoV-2 sequence (green arrow) are used to enrich the retrotransposition junction. In this example, the left read mapped to human chromosome 1 (blue). The right read mapped to the 3′-end of a SARS-CoV-2 RNA sequence (pink) starting from the enrichment primer sequence (green arrow), covering a poly-A tract (orange) and ending with a human sequence (blue) at the retrotransposition site containing a LINE1 endonuclease recognition sequence (TTTT/A, purple).

**Figure 2 viruses-15-00629-f002:**
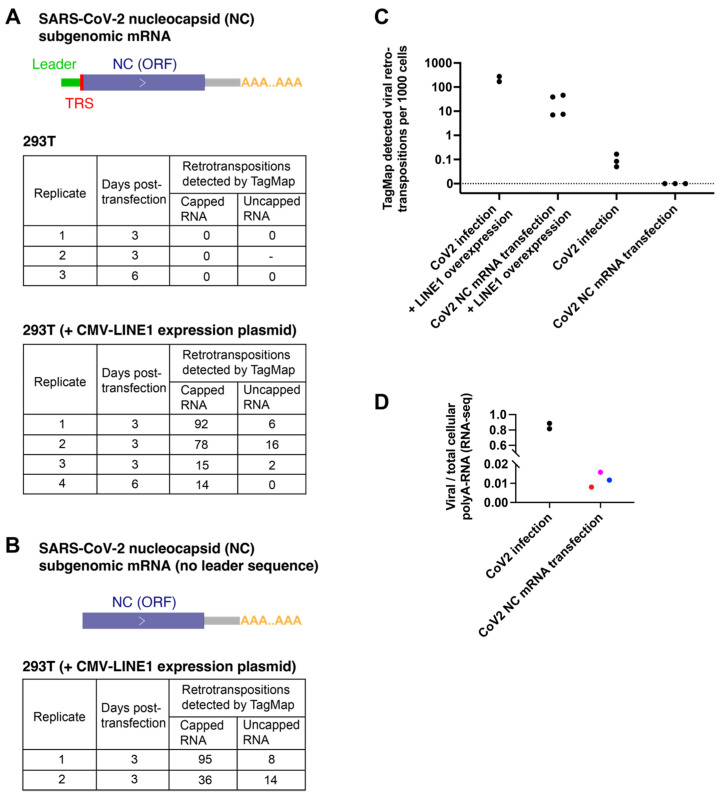
No retrotransposition events were detected by TagMap in cells transfected by viral nucleocapsid mRNA alone. (**A**,**B**) Retrotranspositions detected by TagMap in 293T cells transfected with SARS-CoV-2 nucleocapsid (NC) subgenomic mRNA (**A**) with the leader sequence or (**B**) without the leader sequence, with or without LINE1 overexpression. 293T cells were transfected by the listed NC mRNA with a concentration of 1 μg RNA per 1mL cell culture medium. For LINE1 overexpression, a CMV-LINE1 plasmid was co-transfected with the NC mRNA, with a concentration of 0.5 μg or 1 μg plasmid per 1 mL cell culture medium. The in vitro transcribed NC mRNA can either express or not express the encoded NC protein depending on whether or not the mRNA was 5′-capped, as listed in the tables. (**C**) Retrotransposition detected by TagMap in cells infected with SARS-CoV-2 or transfected with NC mRNA, with or without LINE1 overexpression. CoV2 infection + LINE1 overexpression: 678 or 1110 retrotransposition events were detected in ~4000 293T cells (Table 1). CoV2 NC mRNA (5′-capped) transfection + LINE1 overexpression: 14–92 retrotransposition events were detected in ~2000 293T cells (**A**). CoV2 infection: 1–2 retrotransposition events were detected in ~12,000–20,000 hESC-ACE2 or Calu3 cells (Table 1). CoV2 NC mRNA (5′-capped) transfection: No retrotransposition events were detected in ~20,000 293T cells (Figure 2A). (**D**) Fractions of viral poly-A RNA relative to total cellular poly-A RNA in cells infected with SARS-CoV-2 or in cells transfected with SARS-CoV-2 NC mRNA. For SARS-CoV-2 infected cells, Calu3 cell RNA was harvested 2 days post-infection; RNA-seq data were from a previous publication [27]. For cells transfected with NC mRNA, 293T cells cultured in 24-well plates were transfected by 0.5 μg (red dot), 1 μg (magenta dot), or 2 μg (blue dot) RNA per 1mL cell culture medium for 6 h and then lysed for extraction of cellular RNA and poly-A RNA-seq using the same method that was used for infected cells.

**Figure 3 viruses-15-00629-f003:**
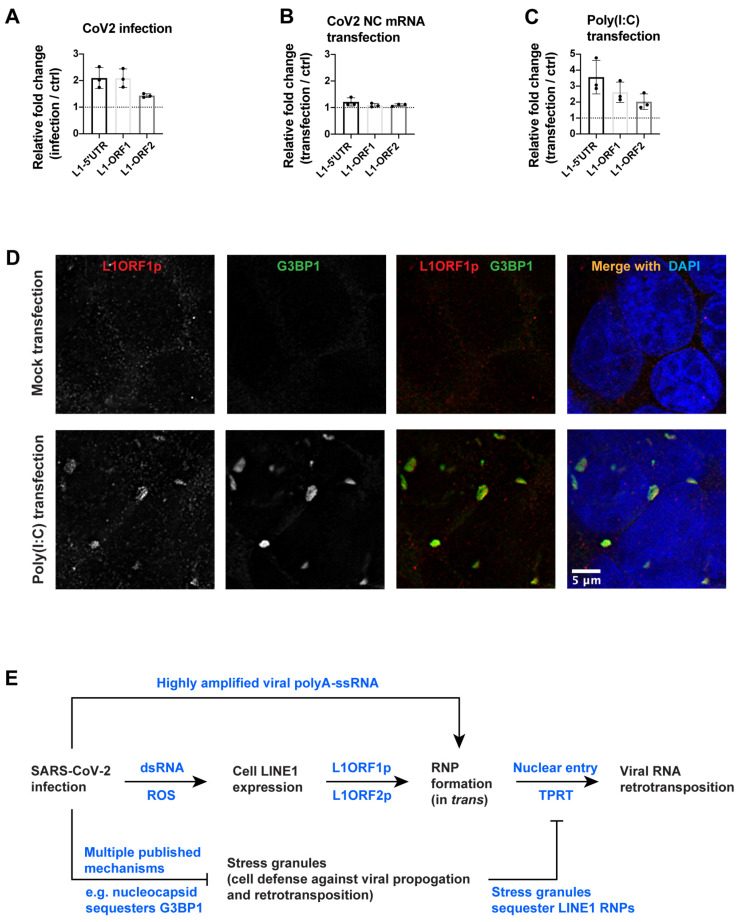
SARS-CoV-2 infection and related cellular stress, but not SARS-CoV-2 nucleocapsid mRNA transfection, can induce endogenous LINE1 expression. (**A**–**C**) LINE1 mRNA level fold-change detected by RT-qPCR in (**A**) Calu3 cells infected by SARS-CoV-2 for 2 days, (**B**) 293T cells transfected by nucleocapsid (NC) mRNA (~1.8 kb with polyA), with 0.5 μg RNA per 1mL cell culture medium for 24 h, or (**C**) 293T cells transfected by a synthetic analog of double-stranded RNA, Poly(I:C) (HMW, 1.5–8 kb), with 2 μg per 1mL cell culture medium for 24 h. RT-qPCR was performed using L1HS/L1PA(2-6) specific primers targeting LINE1 5′-UTR or ORF1, and L1HS specific primers targeting LINE1 ORF2, on purified cellular poly-A RNA (method and primer sequences following the protocols in a previous publication [2], see Materials and Methods). *n* = 3 independent experiments (biological replicates). Data are the mean ± standard deviation (SD). One-tailed *t*-test for LINE1 mRNA upregulation in infected/transfected cells: (**A**) *p* = 0.098 (L1-5′UTR), *p* = 0.021 (L1-ORF1), *p* = 0.303 (L1-ORF2); (**B**) *p* = 0.194 (L1-5′UTR), *p* = 0.379 (L1-ORF1), *p* = 0.315 (L1-ORF2); (**C**) *p* = 0.143 (L1-5′UTR), *p* = 0.121 (L1-ORF1), *p* = 0.020 (L1-ORF2). (**D**) Immunofluorescent staining of L1ORF1p (red) and G3BP1 (green) and merged channels with DAPI staining (blue) in mock-transfected or Poly(I:C) (HMW, 1.5–8 kb) transfected 293T cells. Cells were transfected by Poly(I:C) (HMW, 1.5–8 kb), with 2 μg per 1mL cell culture medium for 24 h. (**E**) A proposed model for mechanisms involved in LINE1-mediated viral RNA retrotransposition in infected cells. In this model, double-stranded viral RNAs (formed by viral replication and transcription) and reactive oxygen species produced upon SARS-CoV-2 infection can stimulate cell endogenous LINE1 expression. The expressed LINE1 proteins (L1ORF1p and L1ORF2p) can interact with single-stranded viral polyadenylated mRNAs to form ribonucleoprotein complex in *trans*, which enter the cell nucleus and retrotranspose through a well-known TPRT mechanism. The high level of viral mRNAs in host cells during viral infection increases the chance of their interaction with LINE1 proteins. Cell stress granules, a cell antiviral/anti-retrotransposition defense mechanism, can be attenuated/cleared by SARS-CoV-2 infection. dsRNA: double-stranded RNA. ssRNA: single-stranded RNA. ROS: reactive oxygen species. RNP: ribonucleoprotein. TPRT: target-primed reverse transcription.

**Table 1 viruses-15-00629-t001:** Method comparison for detecting and quantifying reverse transcribed and retrotransposed SARS-CoV-2 cDNA in virus-infected cells.

Cell	Experiment Replicate	dPCR Detected Total Viral cDNA Copies Derived from the 3′-end Viral RNA ^1^	WGS (Nanopore for 293T-LINE1 and hESC-ACE2, Illumina for Calu3)			TagMap (Enriching 3′-End Integration Junction)		
			Estimated Sequencing- Covered Cells ^2^	Detected Total Viral cDNA Copies Derived from the 3′-End of viral RNA ^3^	Detected Retrotransposed Viral cDNAs with Poly-A Tract and 3′-Flanking Host Sequence ^4^	Estimated Sequencing- Covered Cells ^5^	Detected Retrotransposed Viral cDNAs with Poly-A tract and 3′-Flanking Host Sequence ^6^	TagMap Recovery Efficiency for Retrotransposed Viral cDNAs ^7^
293T-LINE1 ^8^	1	14,000 per 1000 cells	9	233 (26,000 per 1000 cells)	32 (3600 per 1000 cells)	4000	678 (170 per 1000 cells)	5%
	2	20,000 per 1000 cells				4000	1110 (280 per 1000 cells)	8%
hESC-ACE2 ^9^	1	3.9 per 1000 cells	3	0	0	20,000	1 (0.05 per 1000 cells)	–
Calu3	1	14.1 per 1000 cells	4	0	0	12,000	1 (0.08 per 1000 cells)	–
	2	20.6 per 1000 cells	5	0	0	12,000	2 (0.17 per 1000 cells)	–

^1^.- Using PCR primers specific to a viral sequence located within the 3′-end 500 nt of SARS-CoV-2 genomic/subgenomic RNA sequence (upstream of poly-A). ^2^.- For Nanopore or Illumina WGS, cell number estimation was based on sequencing genome coverage. For 293T cells, DNA samples from two independent infection experiments (biological replicates) were pooled and sequenced by Nanopore; the raw data were published in reference [27]. ^3^.- Reporting unique WGS reads that contain viral sequences mapped to the 3′-end 500 nt of SARS-CoV-2 genomic/subgenomic RNA sequence (upstream of poly-A). Nanopore WGS library was constructed without PCR amplification. ^4^.- Retrotransposition events identified by Nanopore WGS were based on the detection of a 3′-end integration junction showing a 3′-end viral sequence and a poly-A tract ranging from 2–65 bp that was directly linked to a LINE1 endonuclease recognition sequence (e.g., 5′-TTTT/A-3′) in the flanking host sequence; these junctions were published in reference [27]. ^5^.- For TagMap, cell numbers were estimated based on Tn5-tagged DNA amount that was used for PCR enrichments. ^6^. Retrotransposed events by TagMap were based on the detection of chimeric DNA sequence showing a 3′-end viral sequence and a poly-A tract flanked by a host sequence. ^7^.- Comparing TagMap (3′-end junction enrichment) detected retrotransposed viral cDNAs (column 8) versus Nanopore reads containing 3′-flanking host sequence (column 6). ^8^.- 293T cells that were transfected with a CMV-LINE1 expression plasmid. ^9^.- Human embryonic stem cells (hESCs) that were transduced with human ACE2.

## Data Availability

Sequencing data generated in this study have been deposited to the Sequence Read Archive (SRA, https://www.ncbi.nlm.nih.gov/sra) with accession numbers PRJNA759137, PRJNA922912, PRJNA922562, PRJNA922942, PRJNA922529. Published sequencing data analyzed in this study are from reference [27] with SRA accession number PRJNA721333.

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
