# Peer review of "LINE1-Mediated Reverse Transcription and Genomic Integration of SARS-CoV-2 mRNA Detected in Virus-Infected but Not in Viral mRNA-Transfected Cells"

_viruses, 2023, doi:10.3390/v15030629_

Round 1

Reviewer 1 Report

The article, proposed by Zhang and Colleagues, compares the interests and limitations of two deep sequencing methods - WGS method vs TagMap method – in the case of reverse transcription events of specific viral mRNAs during SARS-CoV-2 infection. The authors confirmed that the SARS-CoV-2 mRNAs reverse transcription events are strongly associated with LINE1 overexpression and infection. The work is rigorous and easy to follow even for non-specialists but is more dedicated to NGS users and/or retrotransposon specialists.

Author Response

We thank the reviewer for this positive comment.

Reviewer 2 Report

RNA, such as SARS-CoV-2 can be reverse transcribed and integrated into chromosomal DNA by the endogenous retrotransposon LINE-1 encoded retrotransposition machinery. Such events can be detected by whole genome upon overexpression of LINE-1. In cells not expressing LINE-1, whole genome sequencing is not sufficiently sensitive to detect integration of viral DNA. The authors compare the sensitivities of nanopore whole genome sequencing with an enrichment methodology (TagMAp). TagMap is able to interrogate a much larger number of cells and is therefore much more sensitive for detecting events that occur infrequently in cell populations. It is shown that LINE-1 expression increases retrotransposition events approximately 1000-fold. Interestingly, comparing SARS-CoV-2 infection with RNA transfection, retrotansposed RNA sequences were only found in infected cells, but not transfected cells. It is suggested that the large number of RNA copies per cell in the infection pathway results in cell stress which results in derepression of LINE-1 expression. Considering the increased use RNA-based vaccines ,it is noteworthy that transfection of SARS-CoV-2 nucleocapsid does not result in retrotransposition, at least in the cell lines tested. These are important and interesting results and experiments are carefully controlled the results are clearly presented.

Author Response

(The authors gave the same response as above.)

Reviewer 3 Report

I found the study interesting and the conclusions supported by the data. Also, this work is also an essential follow-up to their initial PNAS paper reporting SARS-CoV-2 cDNA insertions that caused significant controversy in the field.

- It would be helpful if an L1-RT inhibitor was examined to address the argument that L1-RT is required to obtain SARS-CoV-2 insertions.

- The authors should discuss a recent paper suggesting that individual SARS-CoV-2 proteins inhibit L1-movement.

Author Response

We thank the reviewer for the positive comment and constructive suggestions.

To address the reviewer’s first suggestion, we have added the new Supplementary Figure S4 and the following sentences in the manuscript:
Similarly, in a different cell type, vascular smooth muscle cells (non-contractile, synthetic phenotype) differentiated from the hESC line H1, we detected no retrotransposition events using
TagMap on cells harvested 3 days after NC mRNA (5’-capped) transfection (Figure S4). In the positive controls, 8 retrotransposition events were detected by TagMap when the LINE1
expression plasmid was co-transfected, and 3 retrotransposition events were detected when the co-transfected cells were treated by RT inhibitors AZT and ABC (Figure S4). These RT inhibitors have been reported to be able to block LINE1 reverse transcriptase activities and LINE1 retrotranspositions [49]. Because the number of retrotransposition events that were detected in the RT inhibitor experiments was small, the significance of these data is unclear.
However, in all the experiments that we performed in which NC mRNA was transfected into cells that did not overexpress LINE1, there was no detectable NC RNA retrotransposition.

To address the reviewer’s second suggestion, we have cited the Li et al. 2023 paper and added a discussion:
Regarding LINE1 retrotransposition, it has been reported that LINE1 cis-retrotransposition or trans-mobilization of Alu and SVA elements were increased upon SARS-CoV-2 infection in cultured 293T cells, analyzed by Nanopore WGS [29]. However, others have reported that, in cultured 293T cells, LINE1 cis-retrotransposition activity was inhibited by several SARS-CoV-2 nonstructural proteins, using transfection-based in vitro LINE1 retrotransposition reporter assays [64], suggesting that there were complex viral-host interactions.

Reviewer 4 Report

In this manuscript, Zhang and his colleagues reported their discovery that SARS-CoV-2 could integrate into the human genome through infection of the HEK 293T cells, but the viral mRNA transfection was not capable of achieving this. Moreover, the authors demonstrated that SARS-CoV-2 infected cells cause de-repression of LINE1, which mediates SARS-CoV-2 genomic integration. Overall, the authors’ findings are significant in contributing to our understanding of the impact of SARS-CoV-2 on the human genome and have implications for the design of safer mRNA vaccines. However, I believe some explanations and experiments need to be done to better support the conclusions reached by the authors.

Major concerns:

The authors claim that SARS-CoV-2 infection induces oxidative stress in the cells, and oxidative stress stimulates LINE1 expression. This is an important part of this manuscript in terms of giving a mechanism for SARS-CoV-2 genomic integration in LINE1 non-overexpressing cells, and the authors should explore more. For example, even though the authors detected a modest increase of LINE1 expression upon SARS-CoV-2 infection, it is still unclear whether this is the only reason to have SARS-CoV-2 retrotranspositions since oxidative stress can change many aspects of cells. The authors should verify the LINE1 function on SARS-CoV-2 retrotranspositions by knockdown LINE1 to see if SARS-CoV-2 genomic integration will be disrupted.

Minor concerns:

1.     In figure 2A-B, can the authors verify the amount of capped and uncapped mRNA after transfection in the cells? It would be necessary to ensure the amount of capped mRNA and uncapped mRNA are at the same level in the cells since capped mRNAs tend to be more stable than uncapped mRNAs.

2.     On page11, the sentence “In LINE1-overexpressing 293T cells, retrotransposed copies in the 5’-capped nucleocapsid mRNA transfected cells were ~10 times lower than that in virus infected cells (Figure 2C).” What is the relationship between this sentence and the following results? The authors should provide explanations.

3.     On page11, the authors claimed that the large fraction of SARS-CoV-2 polyadenylated mRNA increases the chance of interaction between LINE1 protein and viral RNA, and the authors gave an example of the difference of viral RNA fraction between infected Calu3 cells and HEK 293T cells. Are there any published results showing that more SARS-CoV-2 cDNA was detected in Calu3 cells than HEK 293T cells after infection? If yes, please state the results and cite the paper(s); if not, then it is necessary to compare the copy number of SARS-CoV-2 cDNA in these two cell lines.

Author Response

We thank the reviewer for the positive comment and constructive suggestions.

Major concerns:
We proposed SARS-CoV-2 induced oxidative stress as one source of stress in virus-infected cells that could lead to endogenous LINE1 expression and retrotransposition of viral mRNA. We agree with the reviewer that oxidative stress can change many aspects of cells. But we are not stating that oxidative stress is “the only reason” for SARS-CoV-2 retrotranspositions. In our
model (Figure 3E), multiple stresses in infected cells plus highly amplified viral mRNA can increase the possibility of induced LINE1 proteins interacting with the viral RNA and facilitate retrotranspositions.
To address the reviewer’s suggestion of LINE1 knockdown experiment, we added new Supplementary Figure S4 and the following sentences in the manuscript:
“Similarly, in a different cell type, vascular smooth muscle cells (non-contractile, synthetic phenotype) differentiated from the hESC line H1, we detected no retrotransposition events using TagMap on cells harvested 3 days after NC mRNA (5’-capped) transfection (Figure S4). In the positive controls, 8 retrotransposition events were detected by TagMap when the LINE1
expression plasmid was co-transfected, and 3 retrotransposition events were detected when the co-transfected cells were treated by RT inhibitors AZT and ABC (Figure S4). These RT inhibitors have been reported to be able to block LINE1 reverse transcriptase activities and LINE1 retrotranspositions [49]. Because the number of retrotransposition events that were detected in the RT inhibitor experiments was small, the significance of these data is unclear.
However, in all the experiments that we performed in which NC mRNA was transfected into cells that did not overexpress LINE1, there was no detectable NC RNA retrotransposition.”

In thinking about the results of the RT inhibitor experiment, it is possible that overexpressing LINE1 makes it more difficult for the RT inhibitors to block the LINE1 mediated generation of the viral retrotransposition events; however, the fact that overexpressing LINE1 increases the amount of the viral cDNA by about 1000X makes it clear that LINE1 can play an important role in the generation of viral cDNA.

Minor concerns:
1. We thank the reviewer for pointing out this important issue. We compared transfected NC RNA amount by RT-qPCR and found no significant difference after transfecting capped versus uncapped NC RNA. We have added the new Supplementary Figure S3C data and the following sentences in the manuscript:
“There was no significant difference in the amount of transfected RNA on 1 day posttransfection in cells transfected with 5’-capped and non-capped nucleocapsid mRNA (Figure S3C), suggesting the difference in TagMap detected retrotransposition events was not related to a change in RNA stability that was a result of 5’-cap.”

2. We moved the sentence to the end of this paragraph and modified it to make the logic clearer:
“This difference in viral RNA level possibly explains that in LINE1-overexpressing cells, retrotransposed copies in the 5’-capped nucleocapsid mRNA transfected cells were ~10 times lower than that in virus infected cells (Figure 2C).”

3. We apologize for the confusion. The goal here is to compare viral RNA level in infected cells versus transfected cells. Calu3 cells were highly permissive to viral infection and replication but not good for transfection. 293T cells are highly transfectable and thus we chose to use 293T cells in our transfection experiments. The conclusion is that in the most permissive cell lines, viral infection can lead to almost 2 orders of magnitude of higher relative viral RNA level than transfection, which may facilitate retrotranspositions (related to the results in the reviewer’s Minor concern #2).

Reviewer 5 Report

Zhang et al report results from analyses of LINE1-mediated retrotransposition of SARS CoV-2  sequences in three human cell lines.  The experiments are intended to address some of the controversy that arose in response to their previous study which reported rare, but detectable retrotransposition in cultured cells that over-expressed LINE1, along with results of analyses of patient samples, which were consistent with such retrotransposition.  The authors two stated goals in this paper were  (1) to compare the sensitivities of whole genome sequencing (WGS) and Tn5-TagMap detection methods to detect LINE1-mediated retrotransposition of the viral sequences  in cultured human cells, and (2) to determine if transfected viral mRNAs could also be retrotransposed by LINE1 in such cells.  The data provided are of significant value, and the implications will be provocative and of general interest.  Following are some observations and comments concerning each stated goal.

Results from studies directed to goal (1) lean heavily on calculations  (and assumptions) of results from analyses of 293T cells that overexpress LINE1, a system also prominent in their previous report (ref 27) .   Using a reasonable criterion for retrotransposition, the investigators identified 32 integrations by WGS, estimated to be 10 to 20% of the viral cDNA generated in these cells as determined by dPCR.  However, the total number of integrations detected by their TagMap methods was considerably higher, 678-1110.  The investigators conclude from these results that even though it is less sensitive than WGS, TagMap is a better method of detecting retrointegrations, because it enriches for host-viral sequences and can include many more cells. Their results support this conclusion concerning the two methods. However, because the investigator’s previous publication was cited so frequently, it was difficult to discern whether the 293T experiments cited in this manuscript were based on new infections or represented new analyses of samples from previous experiments in ref 27.  It would be helpful it this issue was clarified.  

Data are also reported from analyses of DNA for two additional SARS CoV-2 infected human cell lines, in which LINE1 is not overexpressed. One line, Calu3 was also included in their previous paper; the second, comprising hESC cells that express ACE2, is new for these studies.  In the experiments with these two lines approximately one copy of viral cDNA per 50-250 cells was detected, but no retrotransposed viral sequences were found using WGS, and only 1 or 2 by TagMap.  These results indicate that such integrations are extremely rare in these cells, despite the fact that LINE1 expression is induced  (~2-fold in Calu) upon SARS C0V-2 infection. Unfortunately, the detection of only 1 or 2 integrations in these cases puts these data on somewhat shaky grounds.  The investigators should say more about what might give them (and the reader)  confidence in these results. Could they perform a control with cells in which LINE1 expression was silenced?

For experiments directed to goal (2), viral NC mRNAs were prepared by in vitro transcription, with or without a 5’ leader sequence or a 5’cap to allow translation. Using TagMap, the investigators detected an average of 50 integrations in four experiments with 293T cells that overexpressed LINE1 following transfection with translatable 5’capped mRNA,  and ~5-10-fold less with mRNA that was not capped. The absence of the mRNA leader had no effect and no retrotransposition was detected in cells in which LINE1 was not overexpressed.  The investigators conclude, reasonably,  that the viral NC protein facilitates retrotransposition of viral mRNA in LINE1 over-expressing cells and suggest possible mechanisms. The number of retrotranspositions seen  following transfection with mRNA was an order of magnitude lower than found after virus infection, presumably due to the larger amount of mRNA produced in the latter, and perhaps other features. 

Finally, The investigators show that transfection with Poly(IC),  a proposed surrogate for double stranded viral RNA in infected cells, leads to ~ 3-fold increase in LINE1 expression and its co-localization with stress granules in 293T cells.  From these results and the published capacity of SARS CoV-2 to attenuate the antiviral function of stress granules, the investigators suggest a possible pathway for viral mRNA - LINE1 RNP formation leading to retrotransposition.

In the Discussion section the investigators address the possible biological relevance of their findings for natural infections in humans.  Various factors such as stress and inflammation are cited that might induce sufficient induction of LINE1 expression to promote SARS CoV-2 retrotransposition.  Although virus infection itself is cited, it is difficult to determine if the ~2-fold reported by them and others is meaningful by itself.

The significance of the mRNA transfections are also discussed. The possible role of the viral NC protein and other factors in infected cells to retrotransposition in LINE1 overexpressing cells is noted.  The possible relevance to treatment with mRNA vaccines is an important component of the discussion is the paragraph the describes the limitations of their studies.

Author Response

We thank the reviewer for the summary and constructive suggestions about our article.

1.The DNA from infected LINE1-overexpressing 293T cells used for dPCR and TagMap in this study was the same samples as we used in the previous publication for Nanopore WGS (reference [27]) for method comparison. We have modified the related sentences in the manuscript to clarify this issue.

2.We agree with the reviewer’s comment that the detected viral RNA retrotranspositions in LINE1 non-overexpressing cells were rare. The detections in more than one replicates/cell line suggest these rare retrotranspositions can be detected by TagMap. dPCR results suggested reverse-transcribed total viral cDNAs in the LINE1 nonoverexpressing cells were about 1000 times lower than that in LINE1 overexpressing cells (Table 1). Consistent with this, TagMap detected retrotransposed cDNA in the LINE1 non-overexpressing cells is also about 1000 times lower than LINE1 overexpressing cells (Table 1), which gave us further confidence.

3.The reviewer suggested a control with silenced LINE1. In our RNA transfection
experiment, we added new Supplementary Figure S4. The following sentences were added in the manuscript:

“Similarly, in a different cell type, vascular smooth muscle cells (non-contractile,
synthetic phenotype) differentiated from the hESC line H1, we detected no
retrotransposition events using TagMap on cells harvested 3 days after NC mRNA (5’- capped) transfection (Figure S4). In the positive controls, 8 retrotransposition events were detected by TagMap when the LINE1 expression plasmid was co-transfected, and 3 retrotransposition events were detected when the co-transfected cells were treated by RT inhibitors AZT and ABC (Figure S4). These RT inhibitors have been reported to be able to block LINE1 reverse transcriptase activities and LINE1 retrotranspositions [49].
Because the number of retrotransposition events that were detected in the RT inhibitor experiments was small, the significance of these data is unclear. However, in all the experiments that we performed in which NC mRNA was transfected into cells that did not overexpress LINE1, there was no detectable NC RNA retrotransposition.”

As noted in the response to the fourth reviewer, in thinking about the results of the RT inhibitor experiment, it is possible that overexpressing LINE1 makes it more difficult for the RT inhibitors to block the LINE1 mediated generation of the viral retrotransposition events; however, the fact that overexpressing LINE1 increases the amount of the viral cDNA by about 1000X makes it clear that LINE1 can play an important role in the generation of viral cDNA.